# Unconventional Sites for Diagnosis of Leptospirosis in Bovine Anicteric Fetuses

**DOI:** 10.3390/ani13182832

**Published:** 2023-09-06

**Authors:** Luiza Aymée, Maria Isabel Nogueira Di Azevedo, Luiza Reis, Julia Mendes, Fúlvia de Fátima Almeida de Castro, Filipe Anibal Carvalho-Costa, Guilherme Nunes de Souza, Walter Lilenbaum

**Affiliations:** 1Laboratory of Veterinary Bacteriology, Biomedical Institute, Federal Fluminense University, Niterói 24020-141, RJ, Brazil; luizaaymeeps@gmail.com (L.A.);; 2Curso de Medicina Veterinária, Federal University of Juiz de Fora, Juiz de Fora 36036-900, MG, Brazil; 3Laboratory of Epidemiology and Molecular Systematics, Oswaldo Cruz Institute, Rio de Janeiro 21040-900, RJ, Brazil; 4Embrapa Dairy Cattle, Juiz de Fora 36038-330, MG, Brazil

**Keywords:** cattle, abortion, leptospiral infection, *lip*L32, molecular analysis, necropsy, abomasal fluid

## Abstract

**Simple Summary:**

Abortion is one of the main signs of bovine leptospirosis. The necropsy of the fetuses followed by molecular analysis of the tissues is recommended for the diagnosis. In general, the target sites for diagnosis are the kidneys and liver. However, due to the high diversity of leptospiral strains and anatomic lesions caused by those, this may not be valid for the major agents of bovine leptospirosis. In this context, this study aimed to analyze unconventional sites for the presence of leptospiral DNA in bovine anicteric aborted fetuses. Five anicteric fetuses of a dairy herd with seroreactive cows were submitted for necropsy. After conventional PCR of the *lip*L32 gene of multiple organs, leptospiral DNA was identified in the lungs, heart, subcapsular kidney contents, thymus, liver, kidneys, and the abomasal liquid. Only one fetus was positive in the liver and kidney. Leptospirosis would have been misdiagnosed if only kidney/liver samples had been tested. Therefore, we recommend the investigation of multiple organs, beyond the liver and kidneys, especially in fetuses with anicteric lesions.

**Abstract:**

Background: Bovine leptospirosis is an important reproductive disease and abortion is a major sign, leading to economic impacts. Due to its multifactorial etiology, the proper diagnosis of the cause of the abortion is crucial. Necropsy of the fetuses followed by molecular analysis is recommended for diagnosis, and the investigation mainly occurs in the kidneys and liver. This study aimed to analyze unconventional sites for the presence of leptospiral DNA in bovine anicteric aborted fetuses. Methods: Five fetuses of the same herd were received for necropsy and diagnosis. Conventional *lip*L32-PCR was performed in the fetuses’ kidneys, livers, lungs, hearts, spleens, subcapsular kidney content, abomasal fluid, and in the cavity’s hemorrhagic contents. To complete the investigation, the sera of 30 cows of the herd were collected to perform the serologic screening by Microscopic Agglutination Test. In addition, six subfertile non-pregnant cows from the same herd were selected due to their low reproductive performance, and genital samples (uterine fragment and cervicovaginal mucus) and urine were collected for *lip*L32-PCR. PCR-positive samples were submitted to a nested PCR of the *sec*Y gene and intended for sequencing. Results: The herd presented seroreactive animals (11/30, 36.6%), all against the Sejroe serogroup, with titers between 200 and 1600. In necropsy, four fetuses showed hemorrhagic and anicteric lesions, while one fetus had no macroscopic lesions. Regarding molecular analysis, all the fetuses were positive in *lip*L32-PCR and the positive sites were the heart, lungs, subcapsular kidney content, thymus, kidneys, liver, and abomasal fluid. Only one fetus presented positive results in the kidney and liver, while three fetuses were positive in the abomasal fluid. Five of six cows were positive for *lip*L32-PCR, all being positive only in genital samples. Of the fetuses and the cows, seven sequences were obtained and all were identified as *Leptospira interrogans* serogroup Sejroe serovar Hardjoprajitno. Conclusions: In order to improve the diagnosis of leptospirosis in cows, it is recommended to perform a comprehensive analysis of the samples, beyond the kidneys and liver. Thus, we highly encourage testing multiple organs by PCR to investigate abortions suspected of bovine leptospirosis, particularly in anicteric fetuses.

## 1. Introduction

Leptospirosis is a zoonosis caused by spirochetes of the *Leptospira* sp. genus, which infects humans, domestic and wild animals [1]. In bovines, leptospirosis plays an important role as one of the main reproductive diseases, leading to important economic hazards [2,3]. Leptospires can colonize the genital tract and kidneys of bovines and its transmission occurs from direct contact with infected animals or a contaminated environment [4,5]. Due to its wide genetic and serologic variety, leptospires can be divided into different species and serogroups, respectively [6,7]. In this context, the leptospiral strains of the Sejroe serogroup, such as Hardjoprajitno, Hardjobovis, and Guaricura, are adapted to cattle and widely associated with reproductive failures, including embryonic and fetal losses [4,8]. On the other hand, strains harbored by other animal species and not adapted to bovines are considered incidental [4]. Incidental strains such as Pomona and Icterohaemorrhagiae have been associated with abortion in cattle [2,9,10]. While abortions by Sejroe tend to occur more frequently, infections by incidental strains are associated with occasional outbreaks [11].

Abortions are a major clinical sign in bovines [12,13], but it is often just the most visible sign of a disseminated subclinical infection in the herd, where embryonic death seems to be also common, leading to subfertility [5]. In fact, fetal loss may occur weeks after the dam’s infection and may not be associated with any other clinical signs [11,14]. Nonetheless, abortion itself jeopardizes the herd’s productivity, since it decreases milk production, impairs the replacement of the animals, and increases the treatment costs and the number of artificial inseminations necessary to obtain a calf [15]. Although the economic impact may enormously vary with the country and the zootechnical value of the animals, it has been estimated that the impact of the leptospirosis outbreak with abortions in an Argentinean herd was USD 150,000 [16]. In a French dairy herd, the cost of each abortion by leptospirosis reached EUR 2369 [17].

The exact mechanism of the pathogenesis of leptospiral abortion is yet unclear, but it is suggested that leptospires pass through the placenta during maternal leptospiremia, causing fetal infection that often results in death [4,18,19]. In fetuses with leptospirosis, anatomopathological alterations such as edema, jaundice, the presence of dark-red transudates, as well as focal or generalized hemorrhages have been described [18,20]. However, jaundice and extensive hemorrhages are often linked to the incidental disease, particularly infections caused by Icterohaemorrhagiae or Pomona strains, while few or no visible lesions have been reported in infections determined by Sejroe strains [2,20,21]. 

Since different pathogens can lead to abortions in cattle, the proper diagnosis is crucial to implement control measures. The necropsy of aborted fetuses followed by molecular analysis of its tissues and placenta is highly encouraged for the diagnosis of animal leptospirosis [3,22,23]. Detecting bacterial pathogens in fetuses is particularly challenging due to autolysis, which can happen even before the abortion is expelled, interfering with the bacterial load in the organs [12,13,23]. Following the classic presentation of acute leptospirosis, the liver and kidneys are the main organs recommended for sampling [20,24], and the agent has been demonstrated in icteric fetuses infected by incidental strains, mainly those belonging to serogroups Pomona and Icterohaemorrhagiae; however, since a diversity of leptospiral strains may be involved in abortion pathogeny and its macroscopic lesions may largely vary, other sites cannot be neglected in order to provide an accurate diagnosis, particularly in anicteric fetuses. Therefore, this study aimed to analyze unconventional sites for the presence of leptospiral DNA in bovine anicteric aborted fetuses.

## 2. Materials and Methods

The fetuses were received in the Laboratory of Veterinary Bacteriology of Federal Fluminense University for diagnosis. The further procedures for investigation that involved cows were approved by the Ethical Committee and Animal Use of Federal Fluminense University under protocol 9527220222.

### 2.1. Herd

The herd consisted of dairy production, composed of 91 Holstein lactating cows farmed in an intensive production system (compost barn), located in the Zona da Mata region, of Minas Gerais State, Brazil. The herd is certified as brucellosis-free and the cows had never been vaccinated against leptospirosis. The herd presented constant abortions in the last six months and poor reproductive parameters, including many cows with a history of embryonic mortality and consequent estrus repetition. 

### 2.2. Aborted Fetuses

Five fetuses with development ranging from five to nine months were aborted by different cows in six months. The fetuses were kept frozen until necropsy and analysis. The necropsy and macroscopic evaluation were performed, and the organs collected were the lungs, heart, liver, spleen, kidneys, thymus, and uterus. Approximately 2 g of each tissue was deposited in sterile tubes. Abomasal contents, hemorrhagic liquids, and subcapsular kidney content were collected by aspiration with a syringe of 5 mL. All samples were immediately frozen at −20 °C until the DNA extraction. 

### 2.3. Cows

After the abortions and necropsies, our team went to the herd to collect new samples for additional tests. Except for the mother of fetus B, the other cows were culled before our visit. Six cows with reproductive failures such as embryonic mortality and a history of previous abortion and stillbirths were selected for further molecular analysis, as well as the fetus B’s mother, and five other subfertile non-pregnant cows. From those, cervicovaginal mucus (CVM) and uterine fragments were collected [5,25]. Briefly, the animals were submitted to a physical contention, and low epidural anesthesia with lidocaine 2% and epinephrine (Bravet Inc., Rio de Janeiro, Brazil) was applied. After cleaning and asepsis of the perineal region and vulva, CVM was collected in the vaginal fornix, using a cytologic brush (Kolplast, Itupeva, SP, Brazil) attached to a cytologic device (Botupharma^®^, Botucatu, SP, Brazil). Uterine biopsy was conducted with a Yeoman forceps of 55 cm (ABC Instrumentos, São Paulo, Brazil) in a transcervical approach, while the forceps was guided by transrectal palpation. The obtained fragment was of approximately five millimeters and was deposited in sterile tubes and kept frozen (−20 °C) until molecular analysis.

Besides the six cows selected for the genital samples collection, other 24 cows of the herd were randomly selected for blood collection, in order to complete the sampling of 30 animals for serology. Blood samples were obtained with venopunction of the coccygeal vein and its serum was immediately separated and stored frozen (−20 °C) until analysis.

### 2.4. Microscopic Agglutination Test

For the serology screening of the herd, the serum samples from 30 cows were analyzed by Microscopic Agglutination Test (MAT). The serum samples were tested for antibodies against *Leptospira* using an antigen panel composed of 11 reference strains: *Leptospira interrogans* serovars Copenhageni (strain M20), Hardjoprajitno (strain OMS), Pomona (strain Pomona), Icterohaemorrhagiae (strain Verdun), Canicola (strain Hond Utrecht), Bratislava (strain Jez Bratislava), Australis (strain Ballico), Autumnalis (strain Akiyami), Grippotyphosa (strain Moskva V), *Leptospira borgpetersenii* serovar Hardjobovis (strain Sponselee), all originated from Institut Pasteur, Paris, France; and *Leptospira santarosai* serovar Guaricura (strain FV52), a local Sejroe strain of bovine origin, from Federal Fluminense University, Rio de Janeiro, Brazil. Those strains were selected for the antigen panel because they are known to be predominant in cattle in southeast Brazil [26] and represent seven serogroups: Australis, Autumnalis, Icterohaemorrhagiae, Canicola, Sejroe, Pomona, and Grippotyphosa. The MAT was performed according to the protocol recommended by the guidelines of the World Organization for Health [27]. A twofold dilution of the sera was used to perform testing and to estimate the antibody titers, and the highest dilution with agglutination was taken as the result for the tested sample. The cut-off titer of 200 was considered, according to the recommendations for tropical or endemic regions [26].

### 2.5. Molecular Analysis

The DNA extraction from the samples was performed with DNeasy Blood and Tissue Kit (QIAamp, Qiagen, Courtaboeuf, France) following the manufacturer’s recommendations. Molecular analysis proceeded by conventional PCR, targeting the *lip*L32 gene, exclusive of pathogenic leptospires [28]. The PCR was conducted using the primers LipL32_45F (5′-AAGCATTACTTGCGCTGGTG-3′) and LipL32_286R (5′-TTTCAGCCAGAACTCCGATT-3′), which generate a product with 242 pb. The primers were used in a concentration of 0.75 μL, while the other concentrations of the other reagents were 0.2 μL Taq polymerase, 2 μL MgCl_2_, and 1.25 μL dNTP, 1.25 μL Buffer, in a final volume of 12.5 μL. The reaction was conducted with one cycle of initial denaturation at 94 °C for two minutes, followed by 40 cycles of denaturation at 94 °C for 30 s, annealing the primers to 54 °C for 30 s, and one-minute extension at 72 °C and final extension cycle at 72 °C for five minutes [29]. Ultrapure water and DNA from *L. interrogans* serovar Copenhageni (Fiocruz L1-130) were used as reaction negative and positive control, respectively. After amplification, *lip*L32-PCR products were submitted to electrophoresis in a 2% agarose gel and visualized under UV light after gel red staining. 

The positive samples in *lip*L32-PCR were submitted to a nested PCR of the *sec*Y gene [30], initially using the primers secY_outerF (5′-ATGCCGATCATTTTTGCTTC-3′) and secY_outerR (5′-CCGTCCCTTAATTTTAGACTTCTTC-3′). The primers were used in a concentration of 1 μL, while the other concentrations of the other reagents were 0.4 μL Taq polymerase, 1.75 μL MgCl_2_, and 1.5 μL dNTP, 2.5 μL Buffer, in a final volume of 25 μL. The reaction was conducted with one cycle of initial denaturation at 94 °C for five minutes, followed by 40 cycles of denaturation at 94 °C for 30 s, annealing the primers to 54 °C for 30 s, and one-minute extension at 72 °C and final extension cycle at 72 °C for five minutes. The second reaction was performed with the primers secY_inner_F (5′-CCTCAGACGATTATTCAATGGTTATC-3′) and secY_inner_R (5′-AGAAGAGAAGTTCCACCGAATG3′). Those primers were used in a concentration of 1 μL, and the other concentrations of the reagents were 0.3 μL Taq polymerase, 3 μL MgCl_2_, and 2 μL dNTP, 5 μL Buffer, in a final volume of 50 μL. The reaction was conducted with one cycle of initial denaturation at 94 °C for five minutes, followed by 35 cycles of denaturation at 94 °C for 30 s, annealing the primers to 55 °C for 30 s, and one-minute extension at 72 °C and final extension cycle at 72 °C for five minutes. The obtained amplicons were purified using the Wizard^®^ SV Gel Kit and PCR Clean-Up System (Promega, Madison, WI, USA), following the manufacturer’s instructions, and were submitted to sequencing. Sequencing reactions were performed using the Big Dye Terminator v. 3.1 Cycle Sequencing Kit (Applied Biosystems, San Francisco, CA, USA) on a 3100 automatic DNA sequencer according to the manufacturer’s protocol. The sequences analysis was edited and analyzed by Pairwise/Blast/NCBI software, SeqMan v. 7.0, ClustalW v. 1.35 [31], and BioEdit v. 7.0.1 [32]. A maximum likelihood (ML) tree was constructed using the Tamura–Nei model (TN92) in MEGA 11 software [33], as it was determined to be the best-fitting model of DNA substitution using the Bayesian information criterion. The genetic distances were calculated using the TN92 model on MEGA X. The leptospiral species used for constructing the phylogenetic tree were *Leptospira interrogans*, *Leptospira borgpetersenii, Leptospira santarosai, Leptospira noguchii,* and *Leptospira biflexa* serovar Patoc.

## 3. Results

### 3.1. Serology

A total of 11/30 (36.6%) cows were seroreactive and ten (10/30, 33.3%) were against the Sejroe serogroup, with titers ranging from 200 up to 1600. Of the 10 positive cows for Sejroe, four presented anti-*Leptospira* antibodies titers of 200, three presented titers of 400, one of 800, and three cows had titers of 1600. Only one cow (1/30, 3.3%) was seroreactive to a non-Sejroe, presenting a titer of 200 against Icterohaemorrhagiae. 

Of the 30 cows analyzed in the herd screening serology, six were selected for molecular diagnosis of their CVM. Of those, four cows were seroreactive, three positives to Sejroe serogroup (titers of 400, 800, and 1600), and one positive to Icterohaemorrhagiae serogroup (titer of 200).

### 3.2. Macroscopic Lesions of the Aborted Fetuses

Fetus A had five months of development and presented subcutaneous edema and serosanguineous content on the thorax and abdomen. Lungs presented hemorrhagic suffusions, the liver was enlarged with dark areas and the abomasum had bloody content. The right kidney was friable, with the presence of subcapsular liquid. Fetus B had six months of development and presented hemorrhagic suffusions in the lungs and serosanguineous content in the thoracic cavity. Fetus C had approximately eight months of development and presented only mild hemorrhagic lesions in the lungs. Fetus D had nine months and showed the presence of serosanguineous content in the thoracic cavity, hemorrhagic suffusions in the left lung lobes, and the heart’s left auricle. The liver presented slightly coppery areas. Although the kidneys had normal characteristics, hemorrhagic subcapsular content was present. Fetus E had nine months and did not present any macroscopic alterations in the organs. None of the fetuses presented jaundice or icteric lesions. 

### 3.3. Molecular Diagnosis of Pathogenic Leptospira spp. from the Aborted Fetuses

The *lip*L32-PCR showed the presence of leptospiral DNA in different organs (Table 1). Fetus A was positive only in the lung tissue; Fetus B was positive in the lungs, thymus, and abomasal content; Fetus C was positive in the heart, liver, and kidney parenchyma; Fetus D had positive results in the heart, subcapsular kidney content/perirenal liquid, and abomasal content; Fetus E presented positive results only in abomasal content. DNA products of only two samples were successfully amplified with good quality sequences in *sec*Y nested PCR: the liver and kidney of Fetus C. After nucleotide sequencing, Pairwise/Blast/NCBI comparisons with the GenBank secY gene dataset revealed the species identity as *L. interrogans* with a homology > 99% with strains from serogroup Sejroe (Genotype Hardjoprajitno). 

### 3.4. Molecular Diagnosis of Pathogenic Leptospira spp. from Cows

Five out of the six cows selected for molecular analysis were positive in *lip*L32-PCR (Table 2). From that, all cows were positive in CVM, while four of them were simultaneously positive in uterine fragments. None of the cows were positive in urine. Five good-quality sequences were obtained, one of each positive cow, four from the uterus (UT85, UT86, UT87, UT89), and one from CVM (CVM88). After Pairwise/Blast/NCBI comparisons, all five sequences were identified as *L. interrogans* with an identity > 99% with strains from serogroup Sejroe (Genotype Hardjoprajitno), being similar to the sequences obtained from the fetuses. 

Of the five positive cows in CVM *lip*L32-PCR, three were also positive in serology for the Sejroe serogroup, while the other two PCR-positive cows were negative in MAT (Table 2). The only cow that was negative in PCR was also the only animal to be seroreactive against Icterohaemorrhagiae in the herd screening.

### 3.5. Phylogenetic Analysis of Leptospiral Sequences Obtained

Nucleotide sequences were deposited in GenBank with the accession numbers OR338746-OR338752. Phylogenetic analysis based on the ML-TN92 tree, including *sec*Y *Leptospira* spp. sequences from the reproductive tract of bovines, confirmed species identification of sequences from the present study (Fetus C, UT85, UT86, UT87, UT89, CVM88), as they were included in a highly supported clade (bootstrap = 100%) with *L. interrogans* sequences, separated from other common pathogenic *Leptospira* species (Figure 1). Additionally, phylogeny clearly shows the close proximity with strains from the Sejroe serogroup, more specifically from Hardjoprajitno serovars, with a strong homogeneous cluster formed only with sequences from this serogroup (Figure 1). It is also possible to observe a discrete subcluster formed by sequences from the present study and the strain UF24 (gray rectangle in Figure 1). The overall mean genetic similarity of *L. interrogans* Sejroe serogroup sequences was 99% (TN92 = 0.01, SE = 0.00), and, when comparing sequences from the present study with the UF24 sequence, the genetic similarly was 100% (TN92 = 0.00, SE = 0.00).

## 4. Discussion

Abortions are one of the main clinical signs of bovine leptospirosis [4]. In this present study, five aborted fetuses and five subfertile cows of the same dairy herd were diagnosed with leptospiral infection. Since the herd’s serology presented high seroreactivity against Sejroe, including titers as high as 1600, the involvement of this serogroup is clearly indicated, including both chronic (low titers) and new cases (high titers). The presence of acute seroconversion of the dams was already reported, especially in abortions by Sejroe [10]. Nevertheless, since the abortion can occur weeks after the leptospiral infection, the presence of high titers in serology may not always be present [11,18,34]. In that context, it is known that serology results are only reliable at the herd level [35], and in this herd, as expected, was a useful screening test. It is noteworthy that three out of the five cows that were positive in PCR were also positive in MAT with high titers, which indicates a new infection. However, the other two PCR-positive cows did not present anti-*Leptospira* antibodies, reinforcing that MAT lacks in diagnosing chronic infections, which are very common in bovine leptospirosis. Due to the limitations of MAT in the detection of chronic carriers, instead of performing serology of only the dam of the aborted fetus, we encourage the sampling of multiple cows, with and without reproductive problems, from the same herd, to understand the background of leptospiral infection. Herd serology provides epidemiological information, and when it is employed as the first step in the diagnosis of abortions, confirming or excluding the first suspect, it helps to recognize the source of infection and to enforce the control measures. 

Besides the herd serology, the individual molecular diagnosis of dams is also recommended to understand the cause of abortions. Unfortunately, we could not access all the dams of the studied fetuses, but other cows with reproductive failures of the same herd subgroup were selected. Almost all (5/6) of the analyzed cows were positive, reinforcing the leptospiral infection in the same subgroup of cows. It is also interesting that those animals were only positive in genital samples, and not in urine samples, characterizing the syndrome named Bovine Genital Leptospirosis [5]. This syndrome is mainly characterized by chronic and silent reproductive manifestations, such as embryonic mortality and estrus repetition [5,25], beyond abortions. In that context, cows infected by leptospires can remain underdiagnosed due to the silent signs, impairing the reproductive parameters of the herd. The identification and treatment of leptospiral carriers is crucial to perform leptospirosis control and avoid environmental contamination and new infections, consequently reducing reproductive losses.

Regarding the fetus analysis, the first step toward diagnosis was the necropsy and analysis of the macroscopic alterations in the organs. Although the main referred lesions of leptospirosis are jaundice and pronounced coppery liver and kidneys [20], none of the fetuses analyzed herein presented those signs. Of the five fetuses studied, four presented extensive hemorrhagic lesions and dark-red fluid, alterations that have been reported in abortions by Sejroe [10,18]. It is interesting that one fetus did not present any macroscopic alterations, even with a positive result in abomasum liquid. In this case, since the gross lesions were absent, it is possible that the abortion could have been caused by a dysfunctional placentary environment due to the leptospiral infection and not by the direct infection by the bacteria. Nonetheless, this case reinforces the need for molecular analysis in abomasal liquid even in fetuses without major gross lesions.

Due to the classic pathogeny of the *Leptospira* spp. infection, the key organs recommended for molecular diagnosis are the liver and kidneys [20,23]. Nonetheless, only one fetus presented positive *lip*L32-PCR in the liver and kidney, while the other four fetuses were negative in those organs. If only those samples were analyzed, as commonly recommended, leptospirosis would have been misdiagnosed. At the same time, unconventional sites such as lungs, heart, thymus, subcapsular kidney content, and abomasal content were positive for the presence of leptospiral DNA. In fact, abomasal content and cardiopulmonary organs were the sites with more positive results. The evidence of DNA of *Leptospira* spp. in the abomasal liquid of a bovine aborted fetus was reported by Otaka and colleagues [36], also in a fetus with negative kidney and liver. Similarly, in an aborted foal, the gastric juice presented positively the presence of DNA of leptospires [22]. The fetal stomach content represents the intake of amniotic fluid [13,21], reflecting the placental environment. For this reason, the use of abomasum content as a choice sample for diagnosis has been suggested to diagnose other abortifacient agents such as *Brucella* sp, *Campylobacter foetus* spp., *Coxiella burnetii*, *Listeria monocytogenes*, and mycotic agents [2]. It is important to highlight that in all fetuses with positive abomasal liquid, the content did not present any visible alterations. Therefore, our results encourage the use of this sample to diagnose leptospirosis, particularly anicteric cases, even when there is no pathologic alteration on the abomasum and/or its content. The cardiopulmonary system was also relevant for the molecular investigation. Four fetuses were positive in the lungs or heart, which may represent the importance of those organs to the diagnosis of leptospirosis in anicteric fetuses. Recently, the DNA of leptospires was also identified in the lungs of bovine fetuses with or without interstitial pneumonia [37]. Other unconventional sites such as subcapsular kidney content/perirenal content and thymus also were positive in *lip*L32-PCR and, therefore, complemented the diagnosis. The presence of leptospires in those different sites highlights that several organs should be collected simultaneously for diagnosis, beyond the kidney and liver. When available, molecular analysis of the placenta has been encouraged as well [10,14,38]. Unfortunately, in the present study, placentas were not available for testing. 

The genetic analysis demonstrated high genetic proximity with sequences from the Sejroe serogroup identified in bovines. Not surprisingly, this high genetic homogenous cluster was formed by sequences from the same geographical region—southeastern Brazil. More specifically, the involved agent in the present study was *L. interrogans*, with 100% homology with the UF24 sequence, which is characterized as serogroup Sejroe genotype Hardjoprajitno, confirming the serology results. This sequence was first identified in the uterus of a slaughtered non-pregnant cow, in which the reproductive history is unknown, from Rio de Janeiro, Brazil [39]. Similar sequences were also identified in the follicular fluid of non-pregnant cows with unknown reproductive data [40,41] and genital samples of subfertile naturally infected cows, also in the same region [25]. Those outcomes suggested that this strain may play a major role in bovine leptospirosis in this particular region. It is noteworthy that the studied herd was housed under the controlled conditions of a compost barn, strengthening the epidemiology of cow-to-cow infection, since this system reduces the access of reservoirs of incidental strains. Nonetheless, the lack of vaccination and investigation of the leptospiral carrier status in the new cows introduced in the herd could explain the origin of leptospirosis.

## 5. Conclusions

Abortions are common in the reproductive syndrome caused by leptospirosis. In the case of Sejroe strains, such as Hardjo, fetuses are often anicteric and with no evident visible lesions. In order to increase the diagnostic sensitivity of those fetuses, it is mandatory to perform a comprehensive analysis of samples, beyond the kidneys and liver, especially abomasal liquid, among other unconventional sites. If only the conventional sites are tested, bovine leptospirosis will most probably be misdiagnosed.

## Figures and Tables

**Figure 1 animals-13-02832-f001:**
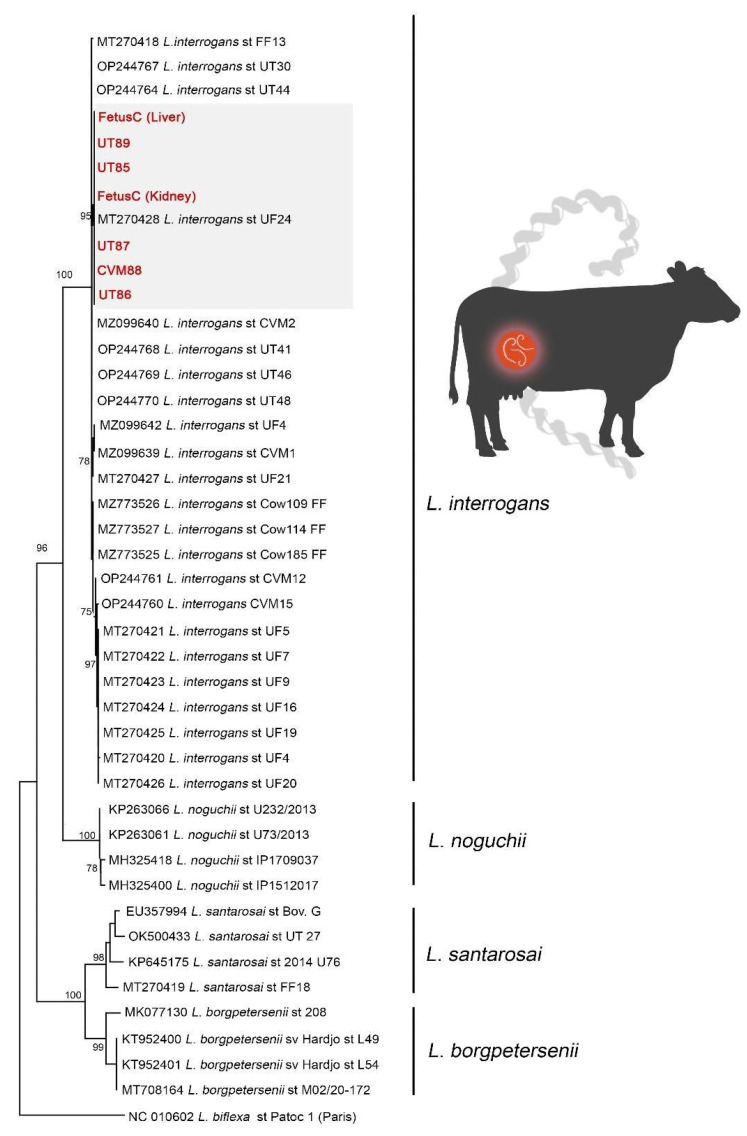
Maximum likelihood phylogenetic tree inferred from partial *sec*Y gene sequences from the present study (red and bold) and GenBank sequences from the main pathogenic *Leptospira* species from the reproductive tract of cows. The subcluster formed by a unique *Leptospira interrogans* Sejroe serogroup haplotype is highlighted in gray. *Leptospira biflexa* serovar Patoc is the outgroup taxa.

**Table 1 animals-13-02832-t001:** Results of conventional PCR targeting *lip*L32 gene in different sites of bovine aborted fetuses.

Animal	Organs	PCR Results
Fetus A (5 months) Female	Lungs	Positive
	Heart	Negative
	Liver	Negative
	Spleen	Negative
	Kidney (parenchyma)	Negative
	Subcapsular kidney content	Negative
	Cavity’s (thoracic and abdominal) hemorrhagic content	Negative
	Abomasal content	Negative
Fetus B (6 months) Female	Lungs	Positive
	Heart	Negative
	Liver	Negative
	Thymus	Positive
	Kidney (parenchyma)	Negative
	Subcapsular kidney content	Negative
	Abomasal content	Positive
	Uterus	Negative
Fetus C (9 months) Female	Lungs	Negative
	Heart	Positive
	Liver	Positive
	Kidney (parenchyma)	Positive
	Subcapsular kidney content	Negative
	Spleen	Negative
	Abomasal content	Negative
	Uterus	Negative
Fetus D (9 months) Female	Lungs	Negative
	Heart	Positive
	Liver	Negative
	Kidney (parenchyma)	Negative
	Subcapsular kidney content	Positive
	Cavity’s (thoracic and abdominal) hemorrhagic content	Negative
	Abomasal content	Positive
	Uterus	Negative
Fetus E (9 months) Male	Lungs	Negative
	Heart	Negative
	Liver	Negative
	Kidney (parenchyma)	Negative
	Subcapsular kidney content	Negative
	Spleen	Negative
	Abomasal content	Positive

**Table 2 animals-13-02832-t002:** Results of *lip*L32-PCR, sequencing, and serology of the six cows with reproductive failures.

Cow	*lip*L32-PCR			Sequencing	Serology
	Uterus	CVM	Urine		
#1 (dam of fetus B)–UT85	Positive	Positive	Negative	*L. interrogans* sg Sejroe	Negative
#2–UT84	Negative	Negative	Negative	-	200 (Ictero *)
#3–UT86	Positive	Positive	Negative	*L. interrogans* sg Sejroe	Negative
#4–UT87	Positive	Positive	Negative	*L. interrogans* sg Sejroe	400 (Sejroe)
#5–CVM 88	Negative	Positive	Negative	*L. interrogans* sg Sejroe	1600 (Sejroe)
#6–UT89	Positive	Positive	Negative	*L. interrogans* sg Sejroe	800 (Sejroe)

CVM—cervicovaginal mucus; sg—serogroup; Ictero *—Icterohaemorrhagiae.

## Data Availability

The data presented in this study are available from the corresponding author on request.

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
