# Peer review of "Unconventional Sites for Diagnosis of Leptospirosis in Bovine Anicteric Fetuses"

_animals, 2023, doi:10.3390/ani13182832_

Round 1
Reviewer 1 Report
Line 154: bibliographic reference 26 is not exhaustive of the description of how the microagglutination test was carried out.
Line 153: you say you have used title 200 as the MAT cut off, but the results reported in the bibliographic reference (26) starting from 100 .
Why did you use 200 as cut off and not 100 ? The 100 is the antibody titre generally used by the scientific community in general as cut off of MAT.
I would like to include some bibliography to support titre 200 and I would also like you to describe in the materials and methods how you carried out the MAT or to cite a more exhaustive bibliography of the method respect to the 26 reference.
Table 2: there is a serology result of 100. Why if you have considered the positives starting from 200. Is it an error? Or the cut off indicated in the methods is an error.
From figure 1 to table 1 there is no connection, in the sense that there is no uniqueness in the identification of positive subjects. In figure 1 or also in its caption it is correct to specify which cows are among those indicated in table 2.
Author Response
Line 154: bibliographic reference 26 is not exhaustive of the description of how the microagglutination test was carried out.
The following sentence and reference were added to the manuscript.
Lines 154-158: “The MAT was performed according to the protocol recommended by the guidelines of the World Organization for Health (30). A two-fold dilution of the sera was used to perform testing and to estimate the antibody titers and the highest dilution with agglutination was taken as the result for the tested sample.”
Line 153: you say you have used title 200 as the MAT cut off, but the results reported in the bibliographic reference (26) starting from 100 .
Why did you use 200 as cut off and not 100? The 100 is the antibody titre generally used by the scientific community in general as cut off of MAT.
I would like to include some bibliography to support titre 200 and I would also like you to describe in the materials and methods how you carried out the MAT or to cite a more exhaustive bibliography of the method respect to the 26 reference.
Dear reviewer, the numbers of the references were incorrect and we profusely apologize for our mistake. In fact, the reference that should be cited in this text is “Pinto, P.daS., Libonati, H., Penna, B., & Lilenbaum, W. A systematic review on the microscopic agglutination test seroepidemiology of bovine leptospirosis in Latin America. Trop Anim Health Prod. 2016;48(2):239-248“. The study recommends the use of the 200 titers in endemic and tropical regions. In general, titers of 100 are useful for non-endemic areas but are not specific enough for endemic tropical regions.
Table 2: there is a serology result of 100. Why if you have considered the positives starting from 200. Is it an error? Or the cut off indicated in the methods is an error.
From figure 1 to table 1 there is no connection, in the sense that there is no uniqueness in the identification of positive subjects. In figure 1 or also in its caption it is correct to specify which cows are among those indicated in table 2.
Thank you very much for your consideration. Indeed, the result presented should be 200 instead of 100, which was corrected in the text. Since our region (southeast Brazil) has a tropical environment and is situated near to Atlantic Forest, which has lots of wild reservoirs, we use the cut-off of 200 in our studies in order to not misdiagnose exposure with an actual infection.
We added more information about the cows in Table 2, to link the table information with the information of the phylogenetic tree.
Reviewer 2 Report
Aymée et al., describe an original article entitled “Unconventional sites for diagnosis of leptospirosis in bovine anicteric fetuses”. The manuscript attempts to bring interesting information about the selection of samples, beyond the kidneys and liver, to perform a more precise and sensitive diagnosis from abortions suspect of bovine leptospirosis. This article is relevant and brings practical newness for the area, beyond be very well written. I believe this manuscript can be published following a minor revision. Please find attached my comments

Author Response
Line 121 and further in the text: please standardize the temperature symbols, add a space between the number and the symbol, use “ºC” not “ºC”.
It was corrected in the text.
Line 146: please change “Hund” by “Hond” Lines 163-164 and further in the text: add a space between the number and the symbols, like “0.6μM” should be “0.6 μM”.
It was corrected in the text.
Lines 172 to 174: the reference provided [29] is a review, I can't find data about the nested PCR there, please provide information about the external primers used, the PCR conditions and volume used, or provide an appropriated reference.
Dear reviewer, we added the following sentences in the text, regarding the protocol used in in our studies.
Lines 178-192: “initially using the primers secY_outerF (5′-ATGCCGATCATTTTTGCTTC-3′) and secY_outerR (5′CCGTCCCTTAATTTTAGACTTCTTC-3′). The primers were used in a concentration of 1 μL, while the other concentrations of the other reagents were 0.4 μL Taq polymerase, 1.75 μL MgCl2, and 1.5 μL dNTP, 2.5 μL Buffer, in a final volume of 25 μL. The reaction was conducted with one cycle of initial denaturation at 94 °C for five minutes, followed by 40 cycles of denaturation at 94 °C for 30 seconds, annealing the primers to 54 °C for 30 seconds, and one-minute extension at 72 °C and final extension cycle at 72 °C for five minutes. The second reaction was performed with the primers secY_inner_F (5′-CCTCAGACGATTATTCAATGGTTATC-3′) and secY_inner_R (5′-AGAAGAGAAGTTCCACCGAATG3′). Those primers were used in a concentration of 1 μL, and the other concentrations of the reagents were 0.3 μL Taq polymerase, 3 μL MgCl2, and 2 μL dNTP, 5 μL Buffer, in a final volume of 50 μL. The reaction was conducted with one cycle of initial denaturation at 94 °C for five minutes, followed by 35 cycles of denaturation at 94 °C for 30 seconds, annealing the primers to 55 °C for 30 seconds, and one-minute extension at 72 °C and final extension cycle at 72 °C for five minutes.”
Lines 180 to 181: Please provide the Leptospira species selected to construct the three.
This sentence was added in lines 202-204 “The leptospiral species used for constructing the phylogenetic tree were Leptospira interrogans, Leptospira borgpetersenii, Leptospira santarosai, Leptospira noguchii, and Leptospira biflexa serovar Patoc”
RESULTS
Lines 203, 2013 and 226: please correct Leptospira gender or species to italic.
Line 239: Table title, the “lip” should be in italic, like “lipL32-PCR”
Sorry for that; It was corrected in the text.
METHODOLOGY, RESULTS, AND DISCUSSION: It’s not clear if the cows that have blood samples collected for MAT analysis were the same that have CVM samples collected to perform molecular analysis. In case positive, the results should be correlated in the discussion section, like for example: is there some relationship between new or chronic cases and the detection of DNA in the samples collected? Did all cows with positive PCR samples also have detectable MAT titers?
All cows that were submitted to molecular analysis were also submitted to serology by MAT, as exemplified in Table 2. In order to reinforce this information, we added these sentences to the results:
Lines 211-214: “Of the 30 cows analyzed in the herd screening serology, six were selected for molecular diagnosis of their CVM. Of those, four cows were seroreactive, three positives to Sejroe serogroup (titers of 400, 800, and 1,600), and one positive to Icterohaemorrhagiae serogroup (titer of 200).”
Lines 246-249: “Of the five positive cows in CVM lipL32-PCR, three were also positive in serology for the Sejroe serogroup, while the other two PCR-positive cows were negative in MAT (Table 2). The only cow that was negative in PCR was also the only animal to be seroreactive against Icterohaemorrhagiae in the herd screening”
We added a better explanation of those results in the discussion as well:
Lines 287-291: “It is noteworthy that three of the five cows that were positive in PCR were also positive in MAT with high titers, which indicates a new infection. However, the other two PCR-positive cows did not present anti-Leptospira spp. antibodies, reinforcing that MAT lacks in diagnosing chronic infections, that are very common in bovine leptospirosis. Due to the limitations of MAT in the detection of chronic carriers, instead of performing serology of only the dam of the aborted fetus, we encourage the sampling of multiple cows, with and without reproductive problems, from the same herd to understand the background of leptospiral infection.”
REFERENCES: gender and species names should be corrected to italic.
It was corrected in the text.
Reviewer 3 Report
General comments :
- First, I want to thanks the authors for this very intersting article that open new way for the diagnosis of leptospirosis.
- analysis or analyzis, the authors have to choose between british or US english
- 5 fetuses is a small number of samples, the conclusions are very interesting but we need a larger scale study to confirm this trend for the sampling. Moreover, the animals are comming from the same herds and then are probably infected with the same strain.
Specific comments :
Line 79-80 : The authors have to specify that the reference [17] is a study performed in French Polynesia and not in France. It's not at all the same conditions.
|
In a French dairy herd, the cost of each abortion by |
79 |
|
leptospirosis reached 2,369 € [17]. |
80 |
Line 112 - 114 : Can the authors specify which analyses (panel of analyses looking for different pathogens like Brucellosis (although the herd is certified free), Listeria, Salmonella, BVDV, IBR, etc.) are performed in case of abortions in this herd?
Are the abortion due to Leptospira if there is no clinical signs ?
|
The herd presented constant abor- |
112 |
|
tions in the last six months and poor reproductive parameters, including many cows with |
113 |
|
a history of embryonic mortality and consequent estrus repetition. |
Line 278-279 : The authors have to verify this citation, it’s very possible it’s wrong, maybe the [16] or the [6] ?
|
Although the main referred lesions |
278 |
|
of leptospirosis are jaundice and pronounced coppery liver and kidneys [20], none of the |
279 |
|
fetuses analyzed herein presented those signs. |
Line 282-285 If no other analyzes have been carried out on the fetuses, is it not premature to conclude on the responsibility of leptospirosis? Perhaps the abortion is due to another cause, infectious or not. Some calves could be born alive and apparently healthy but infected in utero by Leptospira, what do the authors think? Authors need to be more careful about their conclusions.
|
It is interesting that one fetus did not present any macroscopic |
282 |
|
alterations, even with a positive result in abomasum liquid, which demonstrates that abor- |
283 |
|
tions by leptospirosis can also lead to an absence of visible lesions. In addition, it reinforces |
284 |
|
the need for molecular analysis even in fetuses without major gross lesions. |
Author Response
General comments :
- First, I want to thank the authors for this very intersting article that open new way for the diagnosis of leptospirosis.
- analysis or analyzis, the authors have to choose between british or US english
- 5 fetuses is a small number of samples, the conclusions are very interesting but we need a larger scale study to confirm this trend for the sampling. Moreover, the animals are comming from the same herds and then are probably infected with the same strain.
Dear reviewer, we are very thankful for all your consideration and valuable concerns. We have standardized the “analysis” terms in the manuscript. Furthermore, we agree that a greater number of analyzed fetuses would reinforce our results. However, we believe that our findings can induce new larger studies and reduce the misdiagnosis of abortions by leptospirosis.
Specific comments :
Line 79-80: The authors have to specify that the reference [17] is a study performed in French Polynesia and not in France. It's not at all the same conditions.
Dear reviewer, please apologize to us, the reference was incorrect. Unfortunately, in the last edition, the reference list was changed to alphabetical order. We deeply apologize for that. The reference that we were aiming to cite for this information was “Ayral, F. La leptospirose dans les cheptels bovins laitiers en France: impact économique de l'infection. Bulletin des GTV, 2013,69:61-67.”
Line 112 - 114: Can the authors specify which analyses (panel of analyses looking for different pathogens like Brucellosis (although the herd is certified free), Listeria, Salmonella, BVDV, IBR, etc.) are performed in case of abortions in this herd? Are the abortion due to Leptospira if there are no clinical signs?
Indeed, this is an important concern. The studied herd is maintained under controlled conditions and an intensive breeding system. All animals have been submitted to BVDV and IBR serology before their inclusion in this herd. Furthermore, Brucellosis diagnosis is performed annually and there are no signs that evidenced the presence of listeriosis and salmonellosis.
Besides, strong evidence supports the diagnosis of leptospirosis. Necropsy findings presented macroscopic lesions compatible with leptospirosis (in four of the five fetuses); Herd serology positive to Sejroe; PCRpos cows in the herd; reproductive history of the herd comprising estrus repetition, embryonic death, and subfertility.
Line 278-279 : The authors have to verify this citation, it’s very possible it’s wrong, maybe the [16] or the [6] ?
We apologize for the incorrect number of the citation. We have already corrected the number of our references and all of them were double-checked. This information is a citation of Grégoire and collaborators (2020).
Line 282-285 If no other analyzes have been carried out on the fetuses, is it not premature to conclude on the responsibility of leptospirosis? Perhaps the abortion is due to another cause, infectious or not. Some calves could be born alive and apparently healthy but infected in utero by Leptospira, what do the authors think? Authors need to be more careful about their conclusions.
Although there are very interesting options, we do not believe it could apply to that particular herd. It is an intensive breeding facility, and vigilance is constant. Abortions were promptly visualized and all fetuses were dead. Besides, considering the history of the herd, the reproductive symptoms, the lesions and PCR results of the fetuses, and the serology of the herd, we believe we have strong evidence to diagnose leptospirosis.
Regarding fetus E, which did not present any clinical signs and positive lipL32-PCR only in abomasal liquid, it is possible that the fetus's death occurred due to placental alterations and not by the direct infection of the fetus. Based on this, we improved our discussion:
Lines 316-321: “It is interesting that one fetus did not present any macroscopic alterations, even with a positive result in abomasum liquid. In this case, since the gross lesions were absent, it is possible that the abortion could have been caused by a dysfunctional placentary environment due to the leptospiral infection and not by the direct infection by the bacteria. Nonetheless, this case reinforces the need for molecular analysis in abomasal liquid even in fetuses without major gross lesions”
Round 2
Reviewer 1 Report
The work for me can be published as the authors have made the corrections I have requested.